# Myogenin is required for assembly of the transcription machinery on muscle genes during skeletal muscle differentiation

**Abhinav Adhikari**[1,2,3,4,5], **William Kim**[6], **Judith Davie**[1] *

**1** Department of Biochemistry and Molecular Biology and Simmons Cancer Institute, Southern Illinois University School of Medicine, Carbondale, IL, United States of America, **2** Molecular Pathology Unit, Massachusetts General Hospital Research Institute, Charlestown, MA, United States of America, **3** Massachusetts General Hospital Cancer Center, Harvard Medical School, Charlestown, MA, United States of America, **4** Center for Regenerative Medicine, Massachusetts General Hospital, Boston, MA, United States of America, **5** Harvard Stem Cell Institute, Cambridge, MA, United States of America, **6** College of Science, Southern Illinois University, Carbondale, IL, United States of America

* jdavie@siumed.edu

**Data Availability Statement:** All relevant data are within the manuscript and its Supporting Information files.

## Abstract

Skeletal muscle gene expression is governed by the myogenic regulatory family (MRF) which includes MyoD (MYOD1) and myogenin (MYOG). MYOD1 and MYOG are known to regulate an overlapping set of muscle genes, but MYOD1 cannot compensate for the absence of MYOG *in vivo*. *In vitro*, late muscle genes have been shown to be bound by both factors, but require MYOG for activation. The molecular basis for this requirement was unclear. We show here that MYOG is required for the recruitment of TBP and RNAPII to muscle gene promoters, indicating that MYOG is essential in assembling the transcription machinery. Genes regulated by MYOD1 and MYOG include genes required for muscle fusion, *myomake*r and *myomerger*, and we show that *myomaker* is fully dependent on activation by MYOG. We also sought to determine the role of MYOD1 in MYOG dependent gene activation and unexpectedly found that MYOG is required to maintain *Myod1* expression. However, we also found that exogenous MYOD1 was unable to compensate for the loss of *Myog* and activate muscle gene expression. Thus, our results show that MYOD1 and MYOG act in a feed forward loop to maintain each other's expression and also show that it is MYOG, and not MYOD1, that is required to load TBP and activate gene expression on late muscle gene promoters bound by both factors.

## Introduction

Myogenesis, the formation of skeletal muscle, is a multistep process that involves the commitment of mesodermal progenitor cells to myoblasts, expansion of the myoblasts, withdrawal of the myoblasts from the cell cycle and differentiation into myotubes, and fusion to form muscle fibers [1, 2]. The process of skeletal muscle determination and differentiation is controlled by four highly related basic-helix-loop-helix (bHLH) transcription factors known as the myogenic

**Funding:** JKD was supported by National Institute of Arthritis and Musculoskeletal and Skin Diseases of the National Institutes of Health under Award Number RAR068622. https://www.niams.nih.gov/ The funders had no role in the study design, data collection and analysis, decision to publish or preparation of the manuscript.

**Competing interests:** The authors have declared that no competing interests exists.

regulatory factors (MRFs), namely Myf5, MyoD (*Myod1*), myogenin (*Myog*) and MRF4/Myf6 [3–5]. The bHLH domain recognizes the E-box DNA sequence (CANNTG) that gives MRFs sequence specificity in the regulatory regions of the target genes. The MRFs heterodimerize with a member of the ubiquitously expressed E-protein family of bHLH proteins [6].

MYOD1 and MYOG have overlapping targets and bind to very similar E-box DNA sequences and thus, they directly regulate expression of many of the same muscle genes [7, 8]. This previous work has shown that MYOD1 and MYOG appear to have distinctive functions that occur sequentially at individual promoters [7]. It has been proposed that on late differentiation genes bound by both factors, MYOD1 induces chromatin modifications prior to MYOG binding and gene activation [7]. The requirement for MYOG to enhance the expression of genes initiated by MYOD1 has remained unclear.

Deletion of either gene in the mouse has shown that these factors are not functionally redundant. Mice lacking *Myog* die at birth due to a severe reduction of skeletal muscle [9, 10]. *Myod1* was still expressed in these animals [9], but could not compensate for the lack of *Myog* The lack of compensation was also shown in embryonic stem (ES) cells, where ES cells from *Myog*$^{-/-}$ mice had highly attenuated formation of skeletal muscle that could be restored by constitutive expression of MYOG, but not by constitutive expression of MYOD1 [11].

The FACT (FAcilitate Chromatin Transcription) complex is a histone chaperone that has been shown to be crucial for the nucleosome organization during transcription initiation and elongation [12, 13]. The eukaryotic FACT complex comprises two subunits, SSRP1 and SPT16, and both subunits are required for function [12, 14]. The complex is shown to have an affinity for the H2A-H2B histone dimer and thereby could evict or reassemble nucleosomes before and after RNA Polymerase II (RNAPII) during transcription [13, 14]. Mechanistically, in yeast *Saccharomyces cerevisiae*, FACT has been shown to function through histone eviction [15, 16], promoting TATA-box binding protein (TBP) binding to chromatin [17] and its ability to promote transcription elongation [18]. It also was shown that the ubiquitin-proteasomal system fine-tunes FACT activity during transcription elongation [19]. More recently, the FACT complex was shown to recycle histones locally after the passage of RNAP II, consistent with its functional role in nucleosome reassembly during transcription [20].

We have previously shown that MYOG interacts with the FACT complex during the early steps of gene activation during skeletal muscle differentiation [21]. Recruitment of the FACT complex by MYOG then allows FACT to disassemble and reassemble chromatin at muscle genes before and after RNAPII during transcription. We discovered that the FACT complex exclusively interacts with MYOG but not MYOD1 in skeletal muscle, and they do not occupy any muscle gene promoters at the undifferentiated stage where there is no MYOG expression but only MYOD1. Here, we sought to understand the contribution of MYOG on muscle gene promoters bound by MYOD1 and MYOG.

Using a combination of different approaches in 10T1/2 fibroblasts and C2C12 myoblasts, we show that MYOG is required for the recruitment of the transcription initiation machinery to muscle specific genes. The binding of TATA-box binding protein (TBP) and RNAPII are dependent on MYOG. Surprisingly, during differentiation, we also found that MYOD1 expression is inhibited upon the loss or depletion of MYOG, showing that MYOD1 expression is dependent on MYOG. We also show that MYOD1 is the only MRF whose expression decreased with the loss of MYOG. However, restoration of exogenous MYOD1 expression could not rescue muscle gene expression in the absence of MYOG. The data show that MYOG is required to both maintain MYOD1 expression and cooperate with MYOD1 to activate muscle genes.

## Results

### MYOD1 activates muscle genes in 10T1/2 fibroblast cells in a MYOG-dependent manner

10T1/2 fibroblasts are known to lack expression of the myogenic regulatory factors and the introduction of a single MRF can convert the fibroblasts into myocytes, leading to fusion and muscle-specific gene expression under mitogen-deficient differentiation condition [22, 23]. To determine if MYOD1 could activate genes in the absence of MYOG, we first stably overexpressed exogenous MYOD1 (pMyoD) in 10T1/2 fibroblast cells. The pEF6 plasmid in which *Myod1* was cloned was used as a control plasmid (pEmpty). The expression of MYOD1 was confirmed at the level of both mRNA (Fig 1A) and protein (Fig 1B).

The MYOD1 protein expressed from pMyoD contained a C-terminal V5/His tag, resulting in a size shift that could be observed in 10T1/2 cells expressing the epitope tagged MYOD1 with respect to endogenous MYOD1. C2C12 undifferentiated cell extract was used as a positive control as these cells are known to express high levels of MYOD1 (Fig 1B). In agreement with previous studies, the reintroduction of exogenous MYOD1 reactivated myogenic gene expression as shown by the induction of *Myog* (Fig 1B and 1C) and expression of the differentiation specific genes, *Actin 1* (*Acta1*) (Fig 1D) and *Leiomodin 2* (*Lmod2*) (Fig 1E). To determine if MYOG was required for the activation of muscle genes by MYOD1 in 10T1/2 fibroblast cells, we stably transfected a linearized, short-hairpin RNA (shRNA) plasmid targeting *Myog* mRNA (shMyog) in exogenous MYOD1 expressing (pMyoD) 10T1/2 cells. Scrambled (scr) plasmid was used as a control for *Myog* mRNA depletion. The depletion of *Myog* mRNA was confirmed (Fig 1C). We found that expression of the muscle genes, *Acta1* and *Lmod2*, were severely downregulated with the depletion of *Myog* mRNA (Fig 1D and 1E) even though the expression of *Myod1* was unaltered (Fig 1A).

While it was surprising that MYOD1 could not activate *Acta1* and *Lmod2* in the absence of MYOG, these genes are known targets of MYOG. To examine known targets of MYOD1, we also assayed muscle specific genes that had been identified as dependent on MYOD1 and independent of MYOG. These gene included *inhibitor of DNA binding 3* (*Id3*) and *neuronal pentraxin 1* (*Nptx1*, *Np1*) which were identified as transcriptional targets of MYOD1 in growth media (undifferentiated myoblasts) [24]. We also examined a gene identified as a MYOD1 target in an alternative *in vitro* system, *troponin T1*, *slow skeletal type* (*Tnnt1*) [25]. As anticipated, we found that *Tnnt1* (Fig 1F), *Np1* (Fig 1G) and *Id3* (Fig 1H) were activated upon expression of MYOD1. However, we also found that each of the genes were also severely down regulated in the absence of *Myog* (Fig 1F–1H). Thus, even genes that are only activated by MYOD1 appear to also require MYOG for activation.

### Myogenin is required for RNA polymerase II recruitment to muscle genes

In order to better understand the mechanism of how the muscle genes were impaired in activation, we performed chromatin immunoprecipitation (ChIP) assays in 10T1/2 cells expressing MYOD1 and MYOG or only MYOD1. The cell lines used were 10T1/2 cells with the empty vector (EV), pMyoD and scrambled shRNA control (pMyoD+scr) or *Myog* depleted-pMyoD expressing (pMyoD +shMyog) cells. The initial gene promoter chosen for analysis was *Troponin I2*, *Fast Skeletal type* (*Tnni2*). The *Tnni2* promoter was chosen based on a previous study that showed *Tnni2* to be bound by MYOD1 in both myoblasts and myotubes (differentiated) and by MYOG in myotubes [8], suggesting that *Tnni2* is a target of both MYOD1 and MYOG. We have shown that *Tnni2* requires MYOG for activation during embryogenesis [26]. As anticipated, when MYOG was depleted, the enrichment of MYOG was decreased on the

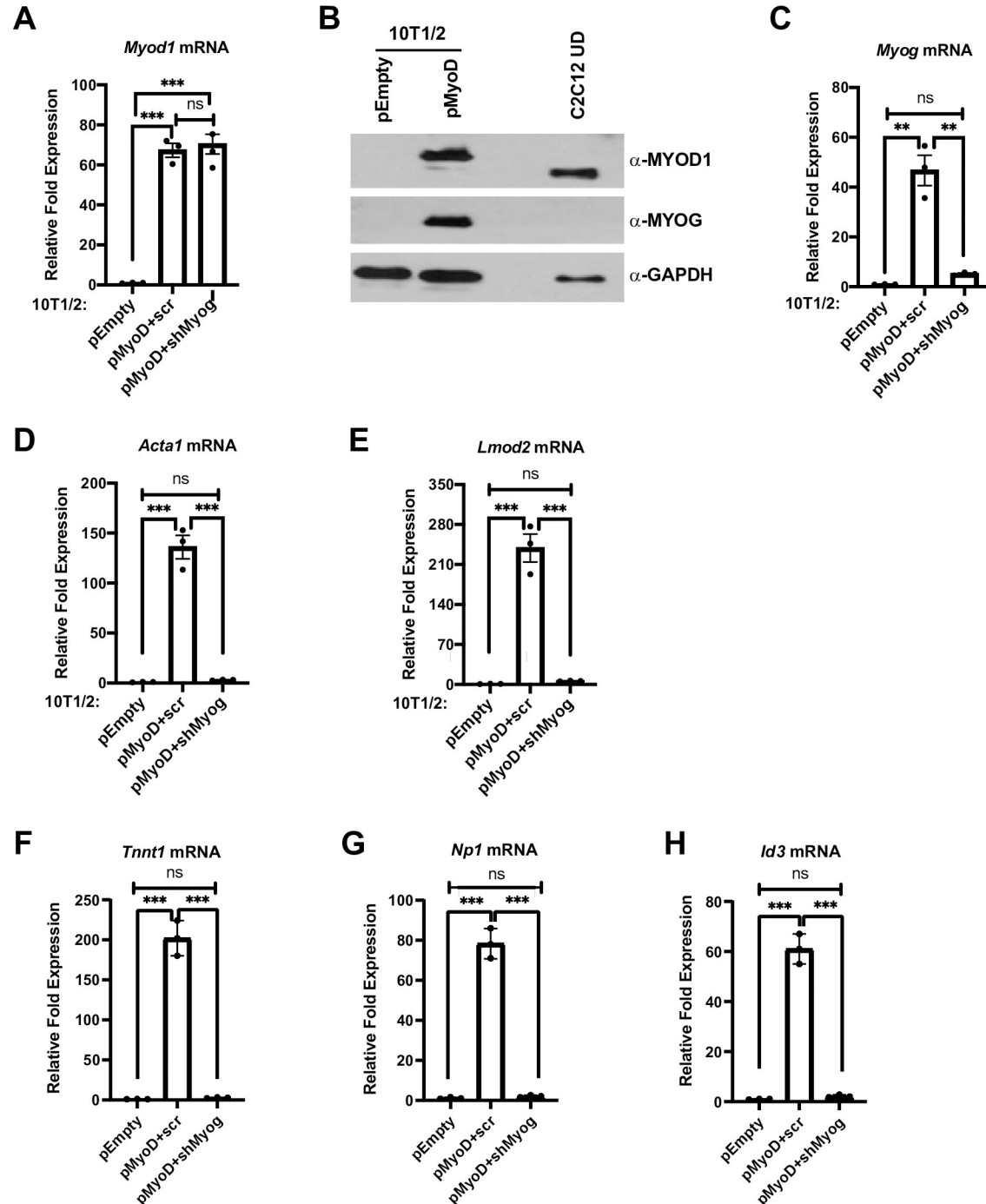

**Fig 1. MYOD1 and MYOG are required to induce muscle gene expression in 10T1/2 fibroblast cells.** A-H. Plasmid expressing MYOD1 (pMyoD) or empty (pEmpty) vector were stably transfected in 10T1/2 cells. Cells were harvested for mRNA and protein and were analyzed for Myod1 by qRT-PCR (A), and western blot (B), respectively. C2C12 UD was used as a positive control for MYOD1. GAPDH was used as a loading control. Stable cell lines overexpressing Myod1 (pMyoD) were further stably transfected with shRNA constructs targeting Myog (pMyoD+shMyog) or scrambled control (pMyoD+scr). Total RNA was extracted and was assayed for *Myod1* (A), *Myog* (C), *Acta1* (D), *Lmod2* (E), *Tnnt1* (F), *Np1* (G) and *Id3* (H). Standard errors (S.E.) from the mean (Mean ± S.E.) represents the error bars. (ANOVA test followed by Tukey's multiple comparison test; ns represents 'not significant', **p<0.01 and ***p<0.001, n = 3 biological replicates).

promoter of *Troponin I2, Fast Skeletal type* (*Tnni2*) (Fig 2A). To confirm this result, we also examined the enrichment of MYOG on the *Leiomodin2* (*Lmod2)* promoter, which we have also shown to be dependent on MYOG during myogenesis [26]. We found that MYOG enrichment was also decreased on the *Lmod2* promoter upon MYOG depletion (Fig 2B). To understand what additional transcription machinery factors might not be recruited in the absence of MYOG, we examined the recruitment of the FACT complex. We have previously shown that MYOG interacts with the histone chaperone FACT complex and the MYOG dependent recruitment of the FACT complex functions in nucleosome disassembly and reassembly at muscle gene promoters during skeletal muscle differentiation [21]. Thus, we assayed for the enrichment of SSRP1, a component of the FACT complex, at the *Tnni2* promoter. We saw a significant decrease in the enrichment of SSRP1 protein at the *Tnni2* promoter (Fig 2C). This result, while in an alternative experimental system, is in agreement with our earlier work [21]. We next examined RNA polymerase II (RNAPII) recruitment and found a significant loss in RNAPII recruitment at the *Tnni2* promoter in the absence of MYOG, indicating defective or decreased transcription (Fig 2D). The loss the RNAPII recruitment could be observed on the *Lmod2* promoter as well (Fig 2E). While these results were in agreement with the severely downregulated mRNA expression of these genes we had observed, the loss of RNAPII recruitment was surprising given that MYOD1 was still present. To confirm that MYOD1 was recruited to these genes, we assayed for MYOD1 enrichment at the *Tnni2* promoter. We found that exogenous MYOD1 was recruited to the *Tnni2* promoter, but this binding was severely reduced upon depletion of MYOG (Fig 2F). We confirmed this result on the *Lmod2* promoter, where we found that MYOD1 was recruited to the *Lmod2* promoter under normal conditions but the recruitment was disrupted upon *Myog* mRNA depletion (Fig 2G). The % input immunoprecipitated by a non-specific antibody control (IgG) at both promoters is shown in Fig 2H and 2I. MYOD1 was lost on these muscle gene promoters upon *Myog* depletion even though the expression of MYOD1 was unaffected (Fig 1A). These data indicate an essential role for MYOG in the assembly of the transcription machinery at muscle genes during skeletal muscle differentiation.

## Myogenin assembles the transcription machinery at muscle gene promoters during skeletal muscle differentiation

We next asked if MYOG was required to assemble the transcription machinery in C2C12 cells, which can undergo skeletal muscle differentiation in cell culture. MYOD1 is highly expressed in C2C12 cells and MYOG expression is induced during differentiation. We assayed for the expression of MYOG by western blot and as anticipated, we found that MYOG was highly expressed after two days of differentiation (D2) (Fig 3A). We then stably depleted *Myog* using shRNA constructs against *Myog* (shMyog) and a scrambled construct (scr) was used as control. The depletion of *Myog* was confirmed at the level of both protein (Fig 3B) and mRNA (Fig 3C). As we had observed in the experiments performed in 10T1/2 cells, the depletion of *Myog* resulted in a significant downregulation in the expression of the muscle genes, *Tnni2*, *Lmod2* and *myosin light chain, phosphorylatable, fast skeletal muscle* (*Mylpf*) (Fig 3C). Expression of the muscle genes was assayed on day 2 of differentiation (D2), when MYOG is normally robustly expressed.

To determine if the recruitment of the transcription machinery to muscle genes would be inhibited upon the depletion of *Myog*, we performed ChIP assays. ChIP assays were performed at two days of differentiation (D2), when MYOG is normally robustly expressed and muscle genes are activated. As anticipated, we saw a decrease in MYOG enrichment on the *Tnni2* promoter upon depletion of *Myog* (Fig 3D) and this result was confirmed at the *Lmod2* promoter

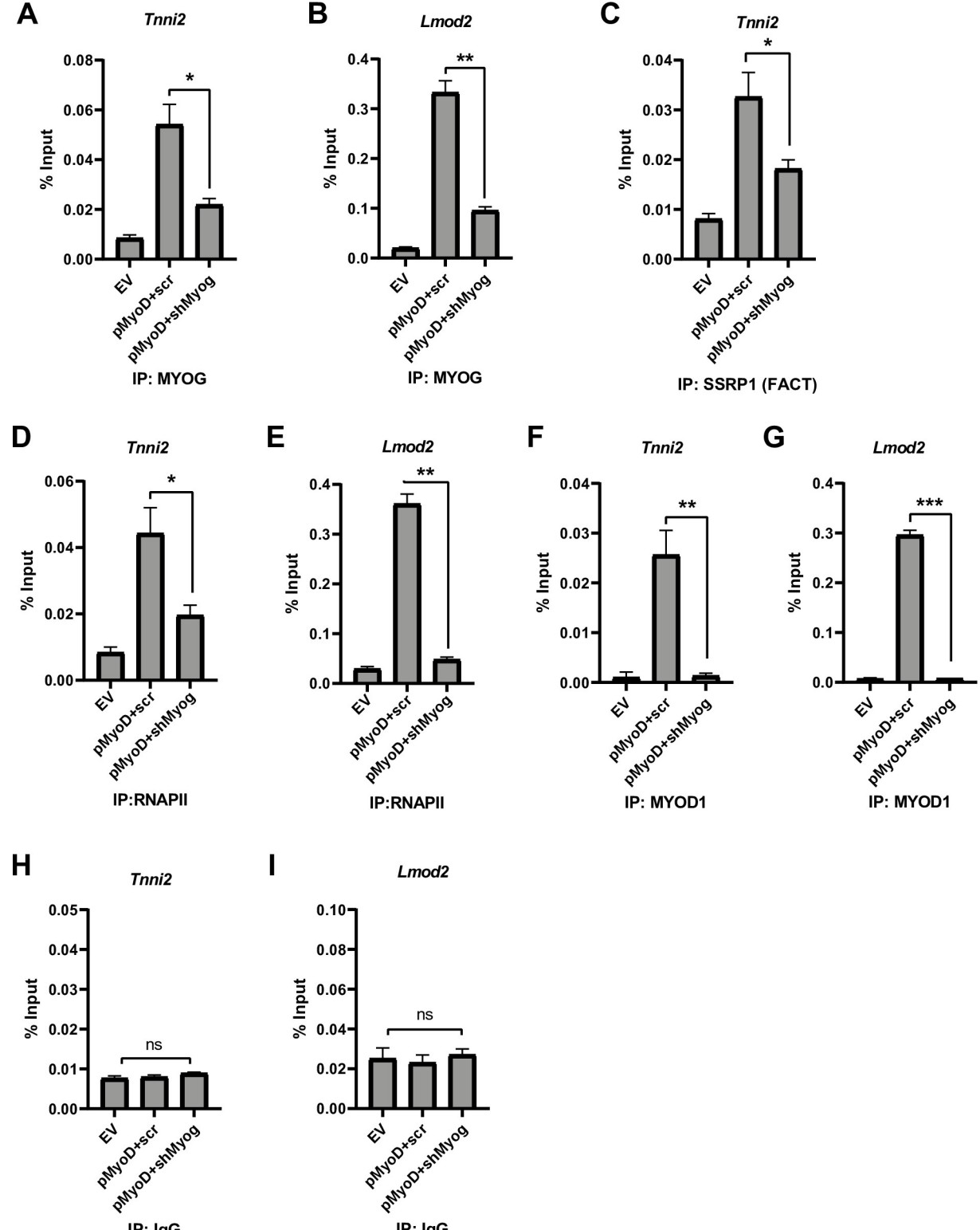

**Fig 2. Depletion of *Myog* impairs RNA polymerase II recruitment to muscle genes.** A-G. Stable 10T1/2 cell lines overexpressing *Myod1* (pMyoD) were further stably transfected with shRNA constructs targeting *Myog* (pMyoD+shMyog) or scrambled control (pMyoD+scr). EV represents empty vector (pEF6/V5 His). ChIP assays were performed with antibodies against MYOG (A and B), SSRP1(C), RNAPII (D and E), MYOD1 (F and G) and IgG (H and I) and primers spanning the *Tnni2* (A, C, D, F and H) and *Lmod2* (B, E, G and I) promoters. Standard

errors (S.E.) from the mean (Mean ± S.E.) represents the error bars. (Student t.test; ns represents 'not significant', *p<0.05, **p<0.01 and ***p<0.001, n = 3 biological replicates).

(Fig 3E). We next examined recruitment of the FACT complex with the depletion of *Myog* on the *Tnni2* promoter. We saw a reduction on the recruitment of both subunits of the FACT complex, SPT16 and SSRP1 (Fig 3F). This result was in concordance with our earlier study (21) and the experiments performed in 10T1/2 cells (Fig 2C). The loss of recruitment of FACT was also observed at the *Lmod2* promoter (Fig 3G). As we had observed in 10T1/2 cells, the recruitment of RNAPII on both the *Tnni2* and *Lmod2* promoters was severely inhibited (Fig 3H). We also assayed for TATA-box binding protein (TBP) enrichment and discovered that the enrichment of TBP was reduced on the *Tnni2* promoter upon the depletion of *Myog* (Fig 3I). To confirm this result, TBP recruitment to the *Lmod2* promoter was also assayed and we found that TBP recruitment to this promoter was inhibited as well (Fig 3J). TBP is a subunit of the eukaryotic general transcription factor, TFIID, which forms the transcription pre-initiation complex [27]. A loss in TBP binding indicates an essential role of MYOG in initiating assembly of the necessary transcription machinery required for transcription initiation at muscle genes during skeletal muscle differentiation.

These results suggest that the loss of muscle gene expression in the absence of MYOG is due to a decrease in the recruitment of RNAPII to muscle specific genes upon *Myog* depletion. To understand why MYOD1 could not compensate for the depletion of *Myog*, we assayed for MYOD1 enrichment on the *Tnni2* promoter. Unexpectedly, we found that MYOD1 enrichment was also decreased upon *Myog* depletion at the *Tnni2* promoter (Fig 3K). To confirm this result, we also examined MYOD1 recruitment to the *Lmod2* promoter and found that MYOD1 recruitment to this promoter was inhibited as well (Fig 3L). These data suggest that MYOD1 recruitment to these muscle genes was partially dependent on MYOG. The decreased recruitment of MYOD1 on these target genes was surprising as many muscle genes have been shown to be regulated by MYOD1 independently of MYOG [7, 8]. As a control for each of these assays, we also examined the % input immunoprecipitated by a non-specific antibody control (IgG) at both promoters (Fig 3M).

The decreased recruitment of MYOD1 was highly surprising, so we next asked if the decrease in MYOD1 enrichment was due to the depletion of *Myog* mRNA or to a global change in MYOD1 expression. To assay for changes in the expression of MYOD1, we first examined expression in normal C2C12 cells in both proliferative (UD) and differentiation conditions (D2). As anticipated, we found that MYOD1 was normally highly expressed during both conditions (Fig 3N). Surprisingly, there was a severe downregulation in the expression of MYOD1 at the level of both mRNA (Fig 3O) and protein (Fig 3P) upon the depletion of *Myog*. These data suggest that MYOG is required to maintain expression of MYOD1 in C2C12 cells during differentiation. This regulation was not observed in 10T1/2 cells, where MYOD1 was expressed from an EF-1α promoter, suggesting that MYOG contributes to activation of the *Myod1* promoter.

## Loss of *myogenin* results in complete loss of differentiation in C2C12 cells

Our results suggested that MYOG was required to maintain the expression of MYOD1. To confirm this result and assure the results were not due to the shRNA approach used, we turned to a CRISPR-Cas9 approach where the *Myog* gene could be disrupted through gene deletion. A CRISPR-Cas9 deletion of *Myog* would allow us to examine MRF dependent gene regulation in the complete absence of MYOG. We designed a guide sequence specific to the first exon of *Myog*, which shares limited homology with the *Myod1* locus. The plasmid containing the

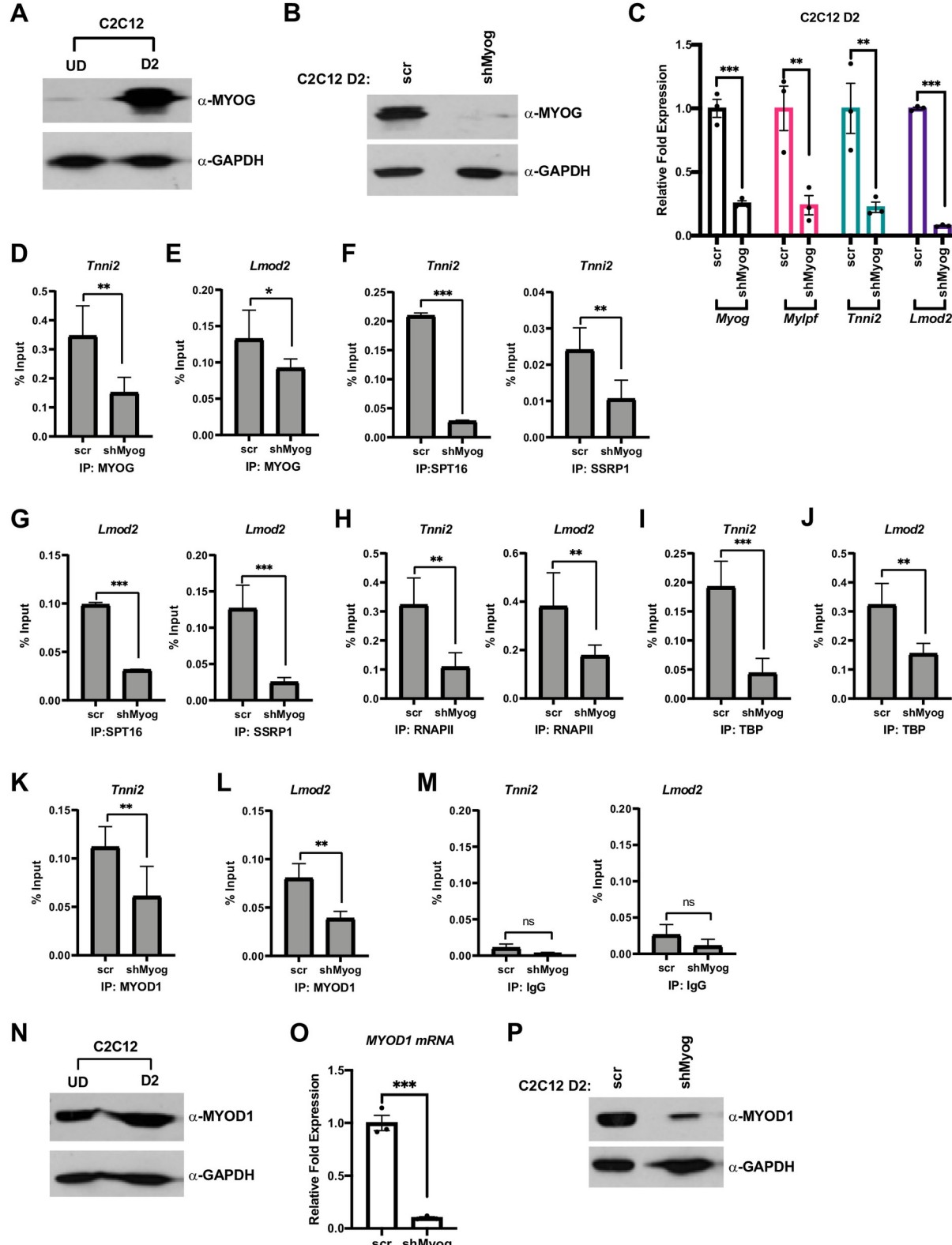

**Fig 3. MYOG assembles the transcription machinery at muscle gene promoters.** A. C2C12 myoblasts were grown in proliferating condition (U.D.) and differentiated for two days (D2) and were assayed for the expression of MYOG by western blot. GAPDH was used as a loading control. B. C2C12 cells were stably transfected with shRNA constructs targeting *Myog* (shMyog) or scrambled control (scr). The selected

clones were differentiated for two days (D2) and were assayed for the expression of MYOG by western blot. GAPDH was used as a loading control. C. Cells as in B were used to extract total RNA, and the mRNA expression of *Myog*, *Mylpf*, *Tnni2* and *Lmod2* were assayed by qRT-PCR. Standard errors (S.E.) from the mean (Mean ± S.E.) represents the error bars. (Student t.test; **p<0.01 and ***p<0.001, n = 3 biological replicates). D-M. ChIP assays were performed on cells as in C with antibodies against MYOG (D and E), SPT16 (F and G), SSRP1 (F and G), RNAPII (H), TBP (I and J) and MYOD1 (K and L), and primers spanning the *Tnni2* and *Lmod2* promoters as labeled. Rabbit IgG (M) was used as a background control. Standard errors (S.E.) from the mean (Mean ± S.E.) represents the error bars. (Student t.test; ns represents 'not significant'. *p<0.05, **p<0.01 and ***p<0.001, n = 3 biological replicates). N. Cells as in A were assayed for MYOD1 protein by western blot. GAPDH was used as a loading control. O-P. Cells as in B were assayed for mRNA and protein expression of *Myod1* by qRT-PCR (O) and western blot (P), respectively. GAPDH was used as a loading control. Standard errors (S.E.) from the mean (Mean ± S.E.) represents the error bars. (Student t.test; ***p<0.001, n = 3 biological replicates).

guide sequence targeting *Myog* was stably transfected in C2C12 cells and stable clones were selected and characterized. Sequencing of the clones revealed that two of the clones (myog K. O. #1 and #2) showed dinucleotide deletion in the *Myog* first exon leading to short transcripts with premature stop codons (Fig 4A). The efficient deletion of the gene was further confirmed by qRT PCR (Fig 4B) and western blot (Fig 4C), which both showed the loss of *Myog* expression.

To confirm that the loss of *Myog* inhibited the differentiation of C2C12 cells, we performed immunofluorescence for myosin heavy chain, a muscle protein commonly used as a marker for skeletal muscle differentiation, and found that the ability of these cells to differentiate was completely lost (Fig 4D).

We next assayed for the expression of additional muscle genes at the mRNA level. Expression of muscle genes known to be dependent on MYOG including *Tnni2*, *Lmod2* and *myosin light chain*, *phosphorylatable*, *fast skeletal muscle* (*Mylpf*), were severely downregulated (Fig 4E). We also examined two genes known to regulate muscle fusion in skeletal muscle, *myomaker* (*Tmem8c*) and *myomerger*, also known as *myomixer* (*Gm7325*). The promoter of *myomaker* contains two highly conserved E box sequences shown to be bound by both MYOG and MYOD1 [28]. We found that both *myomaker* and *myomerger* were severely downregulated by the loss of *Myog* (Fig 4F). These data indicate that expression of both these genes are dependent on MYOG.

We next performed ChIP assays to determine if the transcription machinery assembled at muscle specific promoters with the deletion of *Myog*. The first promoter chosen for analysis was *Tnni2*. As anticipated, MYOG was not recruited (Fig 5A). Again, we found that MYOD1 recruitment was reduced (Fig 5A). We also saw a significant loss in the enrichment of RNAPII and TBP on the *Tnni2* promoter (Fig 5A). The recruitment of the FACT complex was monitored by the recruitment of SSRP1 and we found that SSRP1 recruitment was inhibited by the loss of *Myog* (Fig 5A). Controls for the experiment included a non-specific antibody (IgG) and a histone H3 as a control for total chromatin (Fig 5A). Next, we examined the *Lmod2* promoter. As anticipated, the *Myog* deletion was marked by loss in MYOG recruitment to the promoter of *Lmod2* (Fig 5B). However, at the *Lmod2* promoter, MYOD1 binding was not inhibited (Fig 5B). Even in the presence of MYOD1, we found that RNAPII and TBP recruitment were inhibited (Fig 5B). Recruitment of the FACT complex as detected by *SSRP1* was modestly reduced as well (Fig 5B). Finally, we examined the *Myh3* promoter, as this promoter has been shown to be a late gene that was bound by MYOD1 and MYOG but activated by MYOG [7]. We found that *MYOG* recruitment was lost upon *Myog* deletion, but MYOD1 recruitment was unchanged (Fig 5C). However, like the *Lmod2* promoter, we found that RNAPII recruitment was inhibited, as was recruitment of the FACT complex (Fig 5C). Taken together, the results obtained from the *Myog* depletions largely recapitulated what we had observed with the shRNA approach in both 10T1/2 and C2C12 cells. It was intriguing that in this approach we did observe no loss of MYOD1 recruitment to the *Lmod2* or *Myh3*

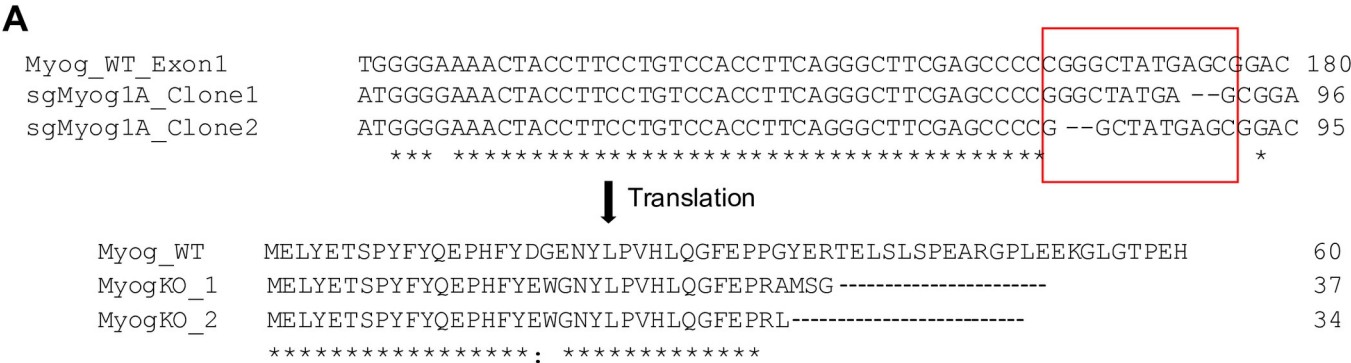

**A**

```
Myog_WT_Exon1     TGGGGAAAACTACCTTCCTGTCCACCTTCAGGGCTTCGAGCCCCCGGGCTATGAGCGGAC 180
sgMyog1A_Clone1   ATGGGGAAAACTACCTTCCTGTCCACCTTCAGGGCTTCGAGCCCCGGGCTATGA--GCGGA 96
sgMyog1A_Clone2   ATGGGGAAAACTACCTTCCTGTCCACCTTCAGGGCTTCGAGCCCCG--GCTATGAGCGGAC 95
                   ***  ****************************************          *
```

↓ Translation

```
Myog_WT    MELYETSPYFYQEPHFYDGENYLPVHLQGFEPPGYERTELSLSPEARGPLEEKGLGTPEH   60
MyogKO_1   MELYETSPYFYQEPHFYEWGNYLPVHLQGFEPRAMSG----------------------   37
MyogKO_2   MELYETSPYFYQEPHFYEWGNYLPVHLQGFEPRL-----------------------   34
           *****************: *************
```

**Fig 4. Deletion of *Myog* impairs muscle differentiation in skeletal muscle.** A. C2C12 cells stably expressing plasmid containing single guide RNA sequence designed against exon-1 of the *myogenin* gene and Cas9 protein were selected, and individual clones were propagated. The *Myog* deletion clones were confirmed by sequencing. The sequences were aligned using Clustal Omega tool with default settings and the indel region is indicated in a red box. B-C. Cells used in A were grown in differentiation condition for two days (D2) and were assayed for the expression of *Myog* at both mRNA (B) and protein (C) by qRT-PCR and western blot, respectively. GAPDH was used as a loading control. Standard errors (S.E.) from the mean (Mean ± S.E.) represents the error bars. (Student t.test; ****p<0.0001, n = 4 biological

replicates). D. Cells as in B were immunostained for myosin heavy chain protein (MHC), a marker for differentiation. DAPI was used to stain the nuclei. Scale bar; 100μm. E-F. Stable C2C12 cells with *Myog* knockout were assayed for the mRNA expression of muscle genes including *Tnni2*, *Lmod2*, *Mylpf* (E), and *Myomaker*, *Myomerger* (F) by qRT-PCR. mRNA expression was assayed at two days of differentiation (D2). Standard errors (S.E.) from the mean (Mean ± S.E.) represents the error bars. (Student t.test; $^{**}$p$<$0.01 and $^{***}$p$<$0.001, n = 3–4 biological replicates).

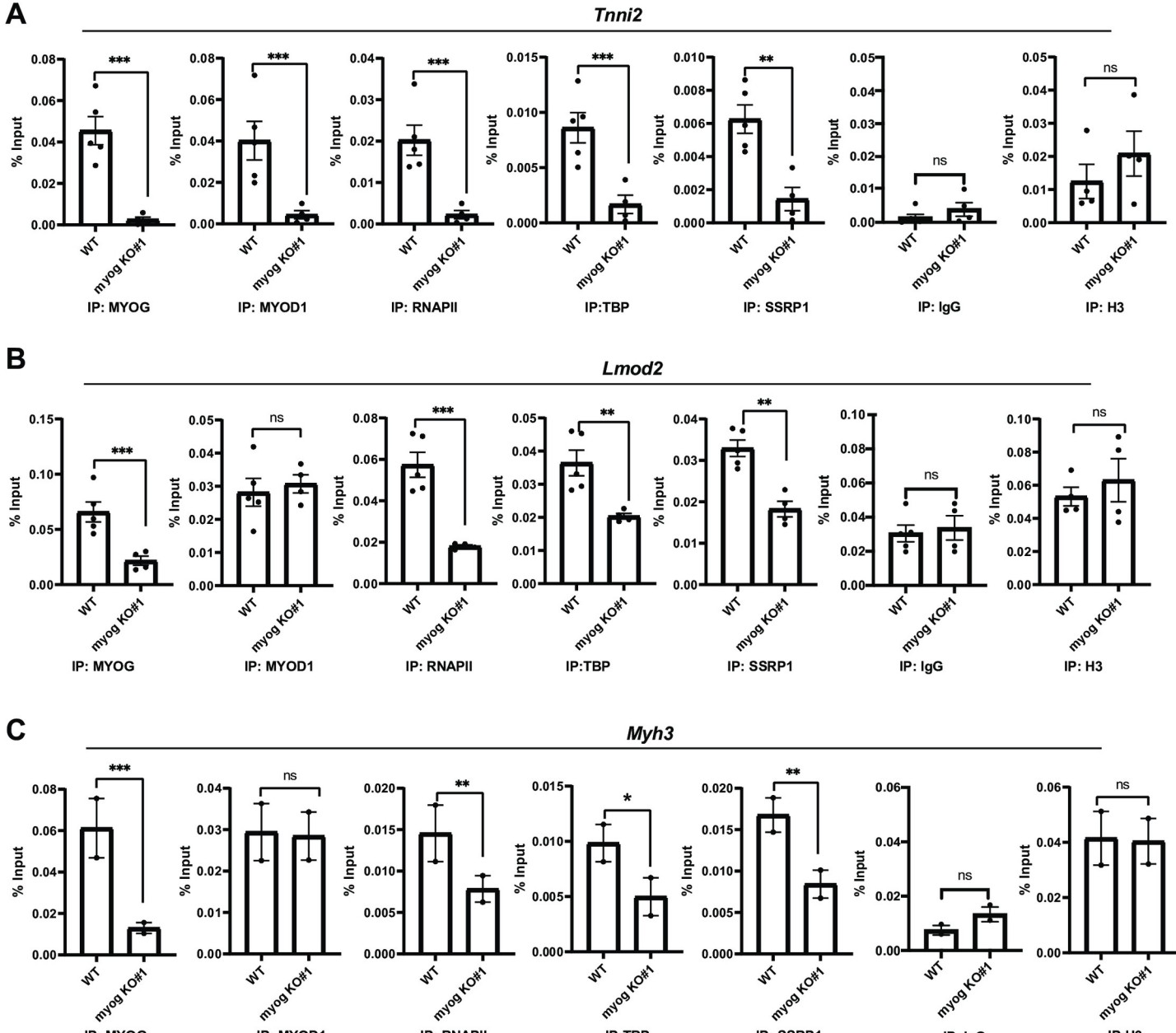

**Fig 5. Deletion of *Myog* impairs transcription assembly at muscle gene promoters.** A-C. ChIP assays were performed on stable C2C12 cells with *Myog* deletion using antibodies against MYOD1, MYOG, SSRP1, TBP, RNAPII and histone 3(H3), and primers spanning *Tnni2*(A), *Lmod2*(B), and *Myh3*(C) promoters. Rabbit IgG was used as a background control. Standard errors (S.E.) from the mean (Mean ± S.E.) represents the error bars. (Student t.test; ns represents 'not significant', $^{*}$p$<$0.05, $^{**}$p$<$0.01 and $^{***}$p$<$0.001, n = 3–5 biological replicates except C where n = 2 biological replicates).

promoters, yet RNAPII recruitment was inhibited, again supporting the requirement for MYOG in assembling the transcription machinery.

Our results throughout this study had indicated that MYOG was required to maintain *Myod1* expression, so we sought to characterize the expression of the MRFs in this system. We found that *Myod1* was downregulated at the level of mRNA (Fig 6A) and protein (Fig 6B) when *Myog* was disrupted in C2C12 cells. These data strongly indicate that MYOG is required to maintain the expression of *Myod1* during differentiation. *Myod1* was the only MRF whose expression was significantly downregulated upon *Myog* deletion (Fig 6A). The expression of *Myf5* and *Myf6* (*MRF4)* were unchanged. A slight decrease in *Myf6* expression was observed in myog KO #2, but the decrease was not statistically significant.

Our results suggested that MYOG was required to assemble the transcription machinery, but the unexpected result that MYOG also maintains *Myod1* expression confounded interpretation of the results. To determine whether the defect in muscle gene expression upon *Myog* deletion was specific to the loss of *Myog* and not due to the subsequent decrease in the expression of *Myod1*, we stably overexpressed *Myod1* (pMyoD) in *Myog* deleted cells. We assayed for the expression of *Myod1* and found that *Myod1* mRNA expression was restored (Fig 6C). As anticipated, *Myog* mRNA expression was not restored as the loss in *Myog* mRNA expression is due to deletion in the *Myog* gene (Fig 6C). We next assayed for muscle gene expression and found that exogenous MYOD1 could not recover the expression of the downstream muscle genes including *Tnni2*, *Lmod2* and *Mylpf* (Fig 6C). At *Tnni2* and *Mylpf*, some reactivation of gene expression was observed, but these results were not statistically significant. No reactivation was observed for *Lmod2*. We next examined *myomaker* and *myomerger* and found that *myomaker* expression appears to be absolutely dependent on the presence of MYOG. For *myomerger*, we observed a statistically significant upregulation of MYOD1 dependent transcription (Fig 6D).

The inability of MYOD1 to rescue most muscle genes was surprising, but the targets chosen for analysis were known targets of MYOG. To examine known targets of MYOD1, we examined the MYOD1 dependent genes used in the 10T1/2 approach. We found that expression of each of these MYOD1 dependent genes was severely down regulated in the absence of *Myog* (Fig 6E). Surprisingly, restoration of MYOD1 did not reactivate any of these targets. Thus, it appears that MYOG and MYOD1 cooperate to activate an overlapping set of gene targets. While we did find that ectopic MYOD1 could activate specific genes such as *myomerger*, it should be noted that there was significant variation in the degree of recue observed so we cannot conclude that MYOD1 fully rescued the transcription deficiency.

Taken together, our results indicate that MYOG has an essential requirement in assembling the transcription machinery on muscle specific genes during skeletal muscle differentiation. MYOD1 and MYOG have significantly overlapping target genes, but there is clearly a requirement for both factors in activating gene expression at promoters. Specific muscle genes appear to have distinct requirements for MYOD1 and MYOG, but our data show that MYOD1 and MYOG work together to activate muscle gene expression, even at targets that MYOG cannot solely activate, such as *Np1*. MYOD1 was known to activate *Myog*, but we show here that MYOG also activates *Myod1* in a feed forward circuit.

## Material and methods

### Cell culture

Proliferating C2C12 myoblast cells (ATCC) and 10T1/2 fibroblast cells (ATCC) were grown in Dulbecco's modified Eagle medium (DMEM) (Hyclone) supplemented with 10% fetal bovine serum (Hyclone) and 1X penicillin (100 I.U./ml)-streptomycin (100μg/ml) antibiotic

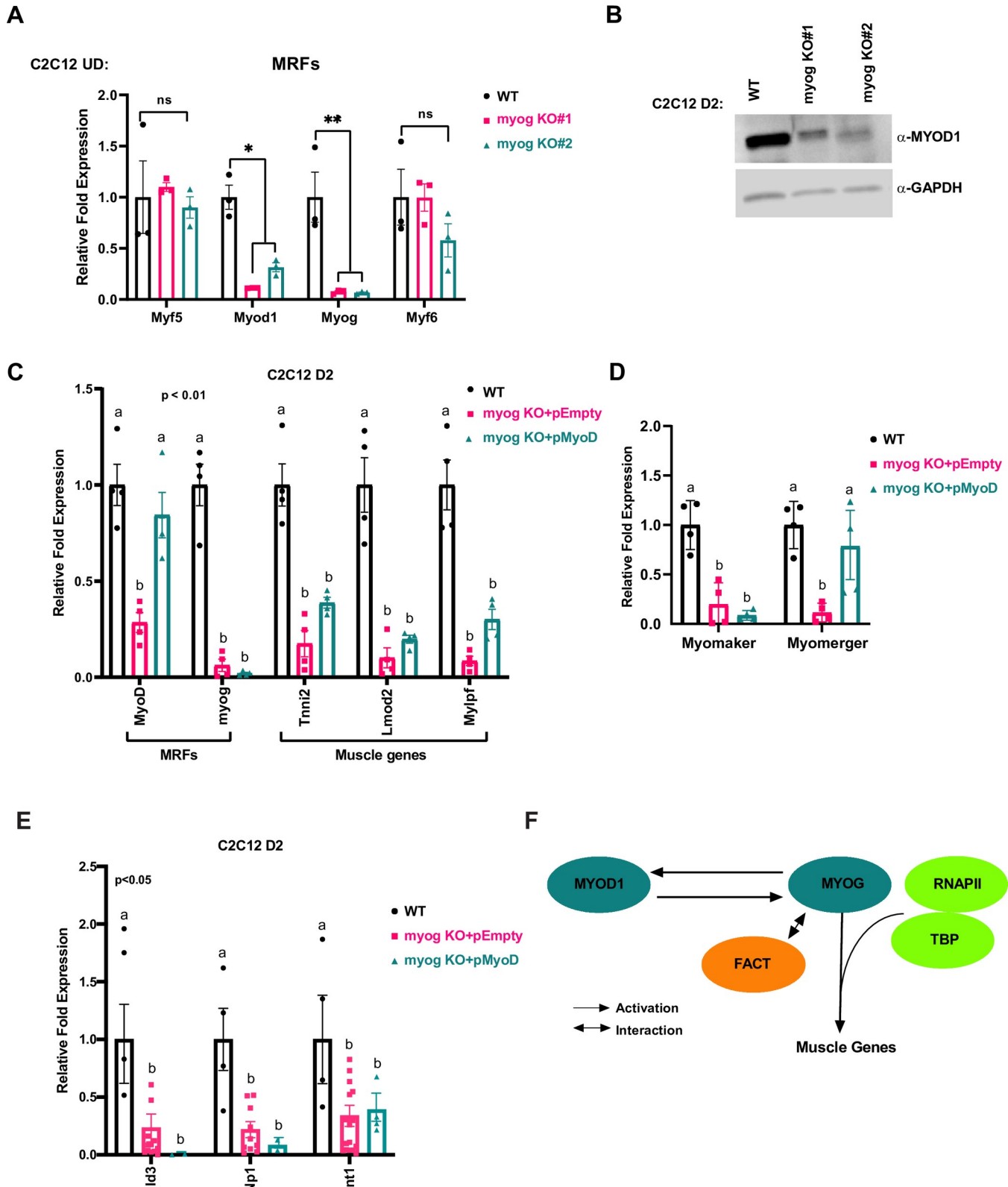

**Fig 6. MYOG regulates MYOD1 in C2C12 cells and is required for muscle gene activation.** A. C2C12 clones with the *Myog* deletion were grown in proliferating condition (U.D.) and were harvested for total RNA and assayed for the expression of *Myf5*, *Myod1*, *Myog*, and *Myf6* by qRT-PCR. Standard errors

(S.E.) from the mean (Mean ± S.E.) represents the error bars. (Student t.test; ns represents 'not significant', *p<0.05 and **p<0.01, n = 3 biological replicates). B. Cells as in A were differentiated for two days (D2) and were assayed for protein expression of MYOD1 by western blot. GAPDH was used as a loading control. C-E. Cells as in A were stably transfected with MYOD1 expression construct (pMyoD) or empty plasmid (pEmpty). The cells were then selected and assayed for the mRNA expression of *Myod1*, *Myog*, *Tnni2*, *Lmod2*, *Mylpf* (C), *Myomaker*, *Myomerger* (D) and the MYOD1 target genes, *Id3*, *Np1* and *Tnnt1* (E) by qRT-PCR. Standard errors (S.E.) from the mean (Mean ± S.E.) represents the error bars. (ANOVA test followed by Tukey's multiple comparisons test for each gene; Samples not sharing an alphabet within a gene group are statistically significant, p<0.01 and n = 3–4 biological replicates). F. A schematic model representing MYOG dependent regulation of muscle genes transcription as well as MYOD1, an upstream regulator of *Myog* expression during skeletal muscle differentiation.

(Corning) in a humidified $CO_2$ incubator at 37˚C according to standard protocols [29]. To induce differentiation of C2C12 myoblasts into myotubes, cells were grown to 80% confluence and the media was switched to DMEM supplemented with 2% horse serum (Hyclone). C2C12 cells were grown in differentiation medium for the number of days indicated in each experiment. All cell lines were authenticated by Bio-Synthesis (Lewisville, TX) using STR analysis on September 14, 2011.

## Plasmids and cloning

The coding sequence of murine *Myod1* was PCR amplified from cDNA reverse transcribed from the total RNA extracted from proliferating C2C12 cells. The PCR amplified *Myod1* insert was cloned into the pEF6/V5 His TOPO TA expression vector according to manufacturer's protocol (Invitrogen). The clones were confirmed by sequencing. Primers used for cloning are listed in S1 Table.

Guide sequences for *myogenin* gene were chosen using an online CRISPR Design Tool (http://tools.genome-engineering.org). Guide containing DNA oligonucleotides were designed and cloned into pSpCas9(BB)-2A-Puro (PX459) V2.0 as described [30]. pSpCas9 (BB)-2A-Puro (PX459) V2.0 was a gift from Feng Zhang (Addgene plasmid # 62987; http://n2t.net/addgene:62987; RRID: Addgene_62987). List of primers are described in S1 Table.

## Cell transfections

C2C12 cells were transfected with TurboFect (Fisher Scientific) or jetPRIME transfection reagent (Polyplus Transfections) according to manufacturer's protocol. All transfections were performed in three technical replicates and a minimum of three biological replicates were performed for each experiment.

## shRNA knock-down and rescue experiment

Short hairpin RNA(shRNA) plasmid for the depletion of *myogenin* (*Myog*) were designed by the RNAi consortium in the pLOK.1 plasmid (Open Biosystems) as previously described [31]. At least three constructs targeting murine *Myog* and one scrambled control were linearized using the *ScaI* restriction enzyme (New England Biolabs), transfected into C2C12 cells or 10T1/2 cells, and the stable colonies are selected with puromycin (2 μg/ml). Individual clones were selected and propagated, and confirmed by mRNA and protein analysis.

For MyoD rescue in C2C12 cells or 10T1/2 stable cell line generation experiment, pEF6-MyoD plasmids transfected in the respective cell lines using Turbofect (Fisher Scientific) transfection reagent as previously described. The stable clones were selected with blasticidin (10μg/ml), clones were confirmed by mRNA and protein analysis.

## CRISPR gene knockout

The plasmid containing the guide sequence targeting *myogenin* was stably transfected in C2C12 cells. Following selection of individual puromycin resistant clones (2μg/ml), genomic

DNA was isolated using Trizol (Invitrogen, ThermoFisher Scientific, Waltham, MA), and PCR amplified targeted regions were sequenced. The gene deletion was also confirmed by sequencing and western blot to assay for protein expression.

## Quantitative real time PCR (qRT-PCR)

RNA was isolated from cells by Trizol extractions (Invitrogen). Following treatment with DNase (Promega), two micrograms of total RNA was reversed transcribed with MultiScribe™ MuLV reverse transcriptase (Applied Biosystems). cDNA equivalent to 40 ng was used for quantitative polymerase chain reaction (qPCR) amplification (Applied Biosystems) with SYBR green PCR master mix (Applied Biosystems). Samples with no reverse transcriptase (no RT) were included for each RNA sample. Realtime PCR (qPCR) data were calculated using the comparative Ct method (Applied Biosystems). Standard deviations from the mean of the ($\Delta$) Ct values were calculated from three independent RNA samples. All the primers used for qRT-PCR described in S1 Table. Where possible, intron spanning primers were used. All quantitative PCR was performed in three technical replicates and in at least three independent RNA samples representing biological replicates were assayed for each time point. For measurements of relative gene expression, a fold change was calculated for each sample pair and then normalized to the fold change observed at HPRT1 and/or 18S rRNA internal reference control.

## Western blot

Cell extracts were made by lysing phosphate buffered saline (PBS) washed cell pellets in radio-immunoprecipitation assay buffer (RIPA) (50mM Tris-HCl pH8.0, 150mM NaCl, 0.1% SDS, 0.5% Na-Deoxycholate, 1% NP-40, 1mM EDTA) supplemented with protease inhibitors (Complete protease inhibitor, Roche Diagnostics). Following incubation on ice for a minimum of 30 min. with vortexing for 10 seconds every 10 minutes, clear lysates were obtained by centrifugation at 4°C. Protein concentrations were determined by Bradford's assay (Bio-Rad). For each sample, 30 μg of protein was loaded on each gel unless mentioned otherwise. Proteins were transferred onto a PVDF membrane using a tank blotter (Bio-Rad). The membranes were then blocked with 5% non-fat milk in 1X TBST (Tris buffered saline plus 1% Tween-20) and incubated with primary antibody overnight at 4°C. Membranes were then washed thrice (3X) 10 minutes each with 1X TBST and incubated with the corresponding secondary antibody in 5% non-fat milk in 1X TBST. Membranes were again washed 3X, 10 minutes each with 1X TBST, incubated with chemiluminescent substrate according to manufacturer's protocol (SuperSignal, Pierce) and visualized by autoradiography or iBright FL1500 Imaging System (Thermo Fisher Scientific). The antibodies used include anti-myogenin (F5D, Developmental Studies Hybridoma Bank (DSHB)), anti-MyoD (5.8 A, Santa Cruz Biotechnology (SCBT) and anti-GAPDH (Millipore). The experiments were repeated at least three times, and the representative blots are shown.

## Immunofluorescence

C2C12 cells were grown on the coverslips till the indicated time point. Cell were fixed with 3.7% formaldehyde for 15 minutes at RT, washed thrice with 1X PBS, blocked with 10% normal goat serum with 1% NP-40 in PBS for 1 hour at RT, and washed three times with PBS. The primary antibodies against myosin heavy chain (1:100, MF-20, DSHB) were incubated overnight at 4°C, washed thrice with PBS, and detected by AlexaFluor-488 goat anti-mouse secondary antibody (1:500, Invitrogen). The cell nuclei were stained with DAPI (300nM, Invitrogen) for 5 minutes at RT. All procedure from secondary antibody staining was

## Chromatin immunoprecipitation assays (ChIP)

ChIP assays were performed as described previously [32]. Briefly, the cells were grown or differentiated for a desired number of days and crosslinked with 1% formaldehyde for 15 minutes at 37°C. The crosslinking reaction was quenched by adding glycine to a final concentration of 0.125M for 5 minutes. The cells were washed in cold 1X PBS thrice, scraped and centrifuged. The cell pellets were then resuspended in cell lysis buffer (5mM PIPES pH 8.0, 85mM KCl, 0.5% NP-40) with protease inhibitors (Complete protease inhibitor, Roche Diagnostics), homogenized with Dounce homogenizer and centrifuged at 4°C. The nuclear pellets were then resuspended in nuclear lysis buffer (50mM Tris-HCl pH8.0, 10mM EDTA, 1% SDS) with protease inhibitors (Complete protease inhibitor, Roche Diagnostics). The chromatin was sonicated at 30–50% amplitude for 18 seconds (3 seconds pulse) for 10 reps to an average length of approximately 500bp. The samples were then centrifuged at 4°C, diluted in twice the volume of IP dilution buffer (0.01% SDS, 1.1% Triton X-100, 16.7mM Tris-HCl pH 8.0, 167mM NaCl, 1.2mM EDTA) with protease inhibitor (Complete protease inhibitor, Roche Diagnostics), and pre-cleared using Protein A beads (Invitrogen). 2μg of specific antibodies were added and the samples were incubated overnight at 4°C followed by the addition of equal volume of Protein A beads (Invitrogen). The antibodies used for the ChIP assays are anti-MyoD (5.8 °A, SCBT), anti-myogenin (F5D, DSHB), anti-SSRP1 (10D1, Biolegend), anti-SPT16 (8D2, Biolegend), anti-RNA polymerase II (H224, SCBT), anti-Histone H3 (D1H2, Cell Signaling) and anti-TBP (D5C9H, Cell Signaling). Rabbit IgG (SCBT) was used as a non-specific control. The complexes were immunoprecipitated with Protein A beads (Invitrogen). Beads were washed twice with cold RIPA buffer (50mM Tris-HCl pH8.0, 150mM NaCl, 0.1% SDS, 0.5% Na-Deoxycholate, 1% NP-40, 1mM EDTA) and once with LiCl wash buffer (250mM LiCl, 50mM Tris-HCl pH8.0, 1mM EDTA, 1% NP-40, 0.5% Na-Deoxycholate) 10 minutes each. Precipitated chromatin complexes were eluted from the beads using elution buffer (50mM Tris-HCl pH 8.0, 1% SDS, 10mM EDTA) and reverse cross-linked overnight at 65°C. The samples were treated with proteinase K. DNA was purified using phenol-chloroform, and ethanol precipitated with 20μg of glycogen as DNA carrier. Realtime PCR (qPCR) was performed using immunoprecipitated DNA template samples to calculate the enrichment at given region using primers spanning region of interest. Primers are described in S1 Table. The real time PCR was performed in three technical replicates. The results are represented as percentage of IP over input signal (% Input). All ChIP assays shown are representative of at least three individual experiments except Fig 5C *Myh3* locus (n = 2 biological replicates). Therefore, all ChIP plots in Fig 5 are plotted as scatterplots with bar graphs where each dot represents an individual replicate value. Standard error from the mean was calculated and plotted as the error bar.

## Statistics

Data are presented as means ± standard errors (S.E.). The dots on the bar graphs represents individual data points from biological replicates. Statistical comparisons were performed using unpaired two-tailed Student's *t* tests or one-way ANOVA followed by Tukey's HSD (honest significant difference) post-hoc test. For one-way ANOVA analysis, all means not sharing the same letter were statistically significant. Probability value of $<0.05$ was taken to indicate significance. All statistical analyses and graphs were made in GraphPad Prism 8.0 software.

## Discussion

The loss of *Myog* causes lethality in mice due to poor muscle formation [9, 10] and even the postnatal loss of *Myog* leads to reduced body size [33], but the molecular mechanism behind these effects was unclear. MYOG was known to be responsible for the activation of muscle specific genes required for sarcomere assembly and contraction and the loss of expression of these genes results in the loss of mature skeletal of muscle, but the specific requirement for MYOG, a relatively weak transcriptional activator, was not understood. We show here that MYOG is essential for the assembly of the transcription machinery on muscle genes during skeletal muscle differentiation. MYOG was known required for late muscle gene expression [7] and we show here that MYOG is required for the loading of TBP and RNAPII to these promoters. We also show that genes which are activated by MYOD1 and cannot be activated solely by MYOG, also require MYOG for activation and cannot be activated by MYOD1 in the absence of MYOG. We also found that MYOG is required to maintain expression of *Myod1*. MYOD1 was known to activate *Myog*, but we show that this effect is reciprocal. Our results show that MYOG and MYOD1 function together to maintain each other's expression and cooperatively activate muscle gene transcription.

The myogenic regulatory factors (MRFs) have been shown to cross-regulate each other and have a degree of functional redundancy and overlap on the target genes, and yet, have unique roles during embryonic and adult myogenesis. All MRFs are basic helix loop helix (bHLH) transcription factors that share high sequence homology, especially between *Myod1* and *myogenin*, have similar interacting E-proteins and have sequence specificity towards the same E-box sequence (CANNTG). Evolutionarily, the MRFs are thought to be duplicated from the same ancestral gene [34, 35]. Thus, when we initially observed the decrease in the expression of *Myod1* due to shRNA-based depletion of *Myog*, we hypothesized that the loss of *Myod1* expression could potentially arise from cross-reactivity for the shRNA constructs used in the study. Sequence analysis of the best performing shRNA and the sequence of the *Myod1* and *Myog* genes showed that there was some homology between the shRNA sequence and *Myod1*. The modest level of similarity would not be predicted to allow targeting of *Myod1* mRNA, but we could not rule out the possibility, although we had not observed targeting of pMYOD1 in 10T1/2 cells. However, the observed loss of *Myod1* expression upon the CRISPR-Cas9 mediated *Myog* gene deletion indicates that the decrease in expression of *Myod*1 with the shRNA approach was, in fact, an outcome of the regulatory loop that exists between MYOG and MYOD1. The *Tnni2* promoter had been shown to be bound by MYOD1 in both myoblasts and myotubes (differentiated) and by MYOG in myotubes [8] and our ChIP assays are consistent with these data. We found that the *Tnni2* promoter showed a consistent loss of MYOD1 enrichment in all three approaches (Figs 2F, 3K and 5A) indicating that *Tnni2* is indeed a common target of both MYOD1 and MYOG during muscle differentiation. Importantly, the restoration of exogenous MYOD1 did not fully rescue expression of *Tnni2* (Fig 6C), confirming the essential role of MYOG in the regulation of muscle genes during skeletal muscle differentiation. MYOG also may play a role in stabilizing or loading MYOD1 on promoters. In 10T1/2 cells with exogenous MYOD1, we observed that MYOD1 promoter binding was dependent on MYOG (Fig 2F and 2G). This was not fully recapitulated in C2C12 cells with a CRISPR deletion of *Myog*, where we saw that MYOD1 was bound to certain promoters, but TBP and RNAPII were not recruited (Fig 5B and 5C). At the *Tnni2* promoter, MYOG was required for the loading of MYOD1 in C2C12 cells (Fig 5A). Our data suggest that C2C12 cells have additional factors that enable MYOD1 loading at specific promoters in the absence of MYOG.

We have previously shown that recruitment of the FACT complex to muscle gene promoters is dependent on MYOG. MYOG dependent recruitment of FACT is an essential mechanism for nucleosome disassembly and reassembly during RNAPII traversion through the gene

body for the expression of muscle genes during differentiation [21]. Intriguingly, we had found that the FACT complex interacts with MYOG and not with MYOD1. MYOD1 is well known as a master regulator of muscle differentiation, which lies upstream of MYOG in the transcriptional cascade of MRFs. MYOD1 has also been shown to regulate the expression of *Myog* during skeletal muscle differentiation [36]. The FACT complex has been shown to be expressed in undifferentiated, proliferative myoblasts and is downregulated as cells differentiate [21, 37]. It is intriguing that our results suggest that FACT is recruited to muscle specific genes by MYOG during early differentiation time points when there is a robust upregulation of MYOG expression and a decrease in FACT expression. Our evidence suggests that the MYOG dependent recruitment of the FACT complex functions to disassemble chromatin and allow TBP to bind muscle specific promoters. TBP can then recruit RNAPII and activate gene expression during skeletal muscle differentiation.

A recent study has discovered the existence of two alternative histone chaperone proteins in differentiated cells. LeRoy *et al*. recently showed the existence of lens epithelium-derived growth factor (LEDGF) and hepatoma-derived growth factor 2 (HDGF2). HDGF2 has been shown to maintain the expression in differentiated myoblasts and regulate successful transcription elongation through nucleosome depletion [38]. It would be interesting to explore the interactions of HDGF2 with the MRFs, specifically MYOG, and the regulation of muscle genes. The loss of TBP and RNAPII enrichment on muscle gene promoters with the loss of *MYOG*, irrespective of the status of MYOD1 enrichment, indicates that the requirement for MYOG is likely the MYOG-dependent recruitment of the FACT complex or any other as of yet unidentified histone chaperones to muscle specific promoters to load TBP and activate gene expression. FACT has been shown to be required for the loading of TBP in *S. cerevisiae* [18], and our results suggest that this function is conserved in higher eukaryotes.

Together, the work presented here indicates that MYOG regulates skeletal muscle differentiation through at least two mechanisms. First, MYOG regulates the expression of downstream muscle genes through recruitment and assembly of the transcription machinery to the target genes, in part through recruitment of the FACT complex. Second, MYOG regulates its expression through the activation of an upstream regulator, *Myod1*, which encodes a known activator of *Myog* through a feed-back activation loop (summarized in Fig 6F). Our work shows that MYOG maintains the expression of MYOD1 and shows that MYOD1 cannot assemble the transcription machinery at the majority of the muscle gene promoters shown here in the absence of *Myog*. The work reveals the unique function of MYOG in initiating skeletal muscle gene expression, thus offering a molecular explanation for the essential role it plays in the development, function and repair of skeletal muscle.

## Supporting information

**S1 Table.**
(PDF)

**S1 Raw images.**
(TIF)

## Acknowledgments

We thank David Kim for technical assistance with the study.

## Author Contributions

**Conceptualization:** Abhinav Adhikari, Judith Davie.

**Formal analysis:** Abhinav Adhikari.

**Funding acquisition:** Judith Davie.

**Investigation:** Abhinav Adhikari, William Kim.

**Methodology:** Abhinav Adhikari.

**Resources:** Judith Davie.

**Supervision:** Abhinav Adhikari.

**Validation:** Abhinav Adhikari.

**Writing – original draft:** Abhinav Adhikari.

**Writing – review & editing:** Abhinav Adhikari, Judith Davie.

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
