## [Decision Letter · Decision Letter 0]

19 Aug 2020

PONE-D-20-22516

Myogenin is required for assembly of the transcription machinery on muscle genes during skeletal muscle differentiation

PLOS ONE

Dear Dr. Davie,

Thank you for submitting your manuscript to PLOS ONE. After careful consideration, we feel that it has merit but does not fully meet PLOS ONE’s publication criteria as it currently stands. Therefore, we invite you to submit a revised version of the manuscript that addresses the points raised during the review process.

The manuscript has been reviewed by two experts in the field. While both reviewers find this manuscript worthy of publication, they have raised a few very pertinent questions about some of the interpretations, and a need for including a more robust control experiment. 

We look forward to receiving your revised manuscript.

Kind regards,

Chhabi K. Govind, Ph. D.

Academic Editor

PLOS ONE

Journal Requirements:

Additional Editor Comments (if provided):

Reviewers' comments:

Reviewer's Responses to Questions

**Comments to the Author**

1. Is the manuscript technically sound, and do the data support the conclusions?

Reviewer #1: Yes

Reviewer #2: Yes

2. Has the statistical analysis been performed appropriately and rigorously? 

Reviewer #1: Yes

Reviewer #2: Yes

3. Have the authors made all data underlying the findings in their manuscript fully available?

Reviewer #1: Yes

Reviewer #2: Yes

4. Is the manuscript presented in an intelligible fashion and written in standard English?

Reviewer #1: Yes

Reviewer #2: Yes

5. Review Comments to the Author

Reviewer #1: The authors are studying the role of myogenic regulatory factors (MRFs) in activating muscle specific genes. Here they define new roles for the myogenin (MYOG) transcription factor, specifically that MYOG is required for recruitment of TBP and RNAPII to promoters and to maintain expression of another MRF, MyoD. The quality of the works is acceptable and the paper makes a modest contribution to the literature.

Specific Comments

1. I have some concerns about the language used, specifically overly strong conclusions. On page 7 (line 154-155) the authors state “We have previously shown that MYOG recruits the histone chaperone FACT complex to muscle gene promoters.” I read reference 21, and the statement is not true. While they did show that MYOG and FACT subunits can interact, they did not show that this interaction is direct. Similarly, while they did show that MYOG knockdown reduces FACT recruitment, they did not show demonstrate that MYOG was sufficient to recruit FACT, as would be tested by targeting a MYOG-Gal4-DBD fusion protein to some other promoter.

2. Similar concerns about overly strong conclusion on page 25-26: “Our evidence suggests that MYOG recruits the FACT complex to disassemble chromatin and load TBP at muscle specific promoters.” The language later on the page about the requirement for MYOG is likely the MYOG-dependent recruitment of the FACT complex” is acceptable.

3. Page 7 lines 157-158. The authors state that there is “a significant decrease in the enrichment of SSRP1 protein at the Tnni2 promoter (Figure 2C),” and that statement is accurate in that there is significant reduction. My question is what is the significance of the residual FACT ChIP signal when MYOG is knocked down. Is that the level of baseline? To answer that one needs controls, and the IgG controls in Figure 3M are not sufficient. I would like to see the residual FACT ChIP signal in a strain without the pMyoG plasmid, i.e. with the pEmpty vector as control.

4. I have a similar question about the baseline control for the experiment in Figure 3L, which shows the 1MYOG knock down results in reduced MyoD recruitment.

5. Figure 4E. Why is there a higher level of target gene expression in myog KO#1 compared to myog KO#2?

Reviewer #2: In the manuscript “Myogenin is required for assembly of the transcription machinery on muscle genes during skeletal muscle differentiation” Adhikari et al. use a combination of shRNA and CRISPR-Cas9 approaches to assess the contribution of MYOG on gene expression and on factor occupancy at several muscle gene promoters. The main conclusions drawn by the authors are that MYOG is required for expression of several muscle genes and for recruitment of RNAPII, FACT and TBP at gene promoters. Moreover, they show that MYOG is required for skeletal muscle differentiation and that MYOG is required to maintain MYOD1 expression.

Overall, these studies are relatively straightforward and the results are robust and clearly presented. There are a few items that should be more clearly addressed by the authors:

1. The dependency for MYOG on MYOD1 expression is clear in the experiments done in C2C12 cells (Figures 3O and 3P) – however, this relationship is not seen in the experiments done with 10T1/2 cells (Figure 1A). The authors should more clearly emphasize this difference, and offer a possible explanation for it.

2. In the results shown in Figure 2F and 2G, shMyog results in loss of MYOD1 occupancy at the Tnni2 and Lmod2 promoters. This loss would not appear to be due to a decrease in MYOD1 expression since these studies were done in 10T1/2 cells. The authors should include some discussion of how they interpret these results in light of the fact that previous studies have shown that, as stated by the authors, “MYOD1 induces chromatin modifications prior to MYOG binding and gene activation” (see lines 53 and 54 in the text) --- their results suggest that MYOG is actually required for MYOD1 promoter recruitment.

3. Although generally convincing, some of the %IP values in the ChIP assays are very low. For example, SSRP1 binding at Tnni2 in the pMyoD+scr sample in Figure 2C is less than 0.007 – this value does not seem convincingly above background levels as judged by the levels of the IgG control in Figure 3M. Some elaboration by the authors on this would be helpful.

4. Although the CRISP-Cas9 experiments elevate the quality of this work, the reasoning for carrying them out provided by the authors (“…it was possible that our shRNA constructs might target both Myod1 mRNA and Myog mRNA due to the high sequence homology between the factors.”) does not seem to be consistent with the results shown in Figure 1A (i.e., shMyog does not reduce Myod1 mRNA). The authors may want to reformulate the added value of these experiments.

5. More information on the antibodies used in the ChIP assays could be helpful for other investigators (e.g., product number for the anti-Histone H3 and TBP antibodies)

6. PLOS authors have the option to publish the peer review history of their article (what does this mean?). If published, this will include your full peer review and any attached files.

Reviewer #1: No

Reviewer #2: No

---

## [Author Response · Author response to Decision Letter 0]

18 Dec 2020

We thank the reviewers for their helpful and constructive comments on our work. We have answered each of the reviewer’s concerns and feel the manuscript is greatly improved.

1. I have some concerns about the language used, specifically overly strong conclusions. On page 7 (line 154-155) the authors state “We have previously shown that MYOG recruits the histone chaperone FACT complex to muscle gene promoters.” I read reference 21, and the statement is not true. While they did show that MYOG and FACT subunits can interact, they did not show that this interaction is direct. Similarly, while they did show that MYOG knockdown reduces FACT recruitment, they did not show demonstrate that MYOG was sufficient to recruit FACT, as would be tested by targeting a MYOG-Gal4-DBD fusion protein to some other promoter.

The suggested experiment is a valuable suggestion for our future work. We have updated the noted sentences for accuracy with our prior results.

2. Similar concerns about overly strong conclusion on page 25-26: “Our evidence suggests that MYOG recruits the FACT complex to disassemble chromatin and load TBP at muscle specific promoters.” The language later on the page about the requirement for MYOG is likely the MYOG-dependent recruitment of the FACT complex” is acceptable.

We have updated the sentence for accuracy.

3. Page 7 lines 157-158. The authors state that there is “a significant decrease in the enrichment of SSRP1 protein at the Tnni2 promoter (Figure 2C),” and that statement is accurate in that there is significant reduction. My question is what is the significance of the residual FACT ChIP signal when MYOG is knocked down. Is that the level of baseline? To answer that one needs controls, and the IgG controls in Figure 3M are not sufficient. I would like to see the residual FACT ChIP signal in a strain without the pMyoG plasmid, i.e. with the pEmpty vector as control.

In reviewing our data to respond to this important point, we noted that the shown calculations in Figure 2 and Figure 3 were not the same. In Figure 2, we had subtracted the IgG signal from each IP value, whereas in Figure 3, we did not subtract IgG, but rather showed IgG as an independent value. We have reanalyzed the data from each experiment and the ChIP shown in Figure 2 is now shown as in Figure 3. The SSRP1 IP signal in the shMyog cells is slightly above IgG background (~2 fold). Additionally, as suggested by the reviewer, we have also performed the ChIP experiments in the 10T1/2 cell line with the EV control to show the residual signal of the FACT complex in 10T1/2 cells and the data are now included in Figure 2. 

4. I have a similar question about the baseline control for the experiment in Figure 3L, which shows the 1MYOG knock down results in reduced MyoD recruitment.

The signal is modestly above background (~2 fold). We have changed the axis scale on the IgG panels to enable this comparison.

5. Figure 4E. Why is there a higher level of target gene expression in myog KO#1 compared to myog KO#2?

We characterized multiple clones from the CRISPR approach to assure our reported results were not due to off targeting effects. KO#1 and KO#2 were two independent clonal isolates and only the Myog genomic region was sequenced in these clones. It is unknown why KO#1 resulted in higher expression of target gene expression, but it could be due to additional targeting events in either of the clones. 

Reviewer #2: In the manuscript “Myogenin is required for assembly of the transcription machinery on muscle genes during skeletal muscle differentiation” Adhikari et al. use a combination of shRNA and CRISPR-Cas9 approaches to assess the contribution of MYOG on gene expression and on factor occupancy at several muscle gene promoters. The main conclusions drawn by the authors are that MYOG is required for expression of several muscle genes and for recruitment of RNAPII, FACT and TBP at gene promoters. Moreover, they show that MYOG is required for skeletal muscle differentiation and that MYOG is required to maintain MYOD1 expression.

Overall, these studies are relatively straightforward and the results are robust and clearly presented. There are a few items that should be more clearly addressed by the authors:

1. The dependency for MYOG on MYOD1 expression is clear in the experiments done in C2C12 cells (Figures 3O and 3P) – however, this relationship is not seen in the experiments done with 10T1/2 cells (Figure 1A). The authors should more clearly emphasize this difference, and offer a possible explanation for it.

We believe that MYOG activates the Myod1 promoter. MYOD1 expression in Figure 1A is driven by the EF-1ɑ promoter in the pEF vector. This important point has been added to the manuscript. 

2. In the results shown in Figure 2F and 2G, shMyog results in loss of MYOD1 occupancy at the Tnni2 and Lmod2 promoters. This loss would not appear to be due to a decrease in MYOD1 expression since these studies were done in 10T1/2 cells. The authors should include some discussion of how they interpret these results in light of the fact that previous studies have shown that, as stated by the authors, “MYOD1 induces chromatin modifications prior to MYOG binding and gene activation” (see lines 53 and 54 in the text) --- their results suggest that MYOG is actually required for MYOD1 promoter recruitment.

It was striking to us that MYOG does appear to be required for MYOD1 loading in 10T1/2 cells, but not at all promoters in C2C12 cells deleted for Myog. These data suggest that MYOG does play a role in stabilizing MYOD1 at promoters, but that other factors present in a normal muscle cell (represented by C2C12 cells), but not in 10T1/2 cells, also have roles in loading and maintaining MYOD1 at specific promoters in the absence of MYOG. We have added text to the manuscript regarding this point.

3. Although generally convincing, some of the %IP values in the ChIP assays are very low. For example, SSRP1 binding at Tnni2 in the pMyoD+scr sample in Figure 2C is less than 0.007 – this value does not seem convincingly above background levels as judged by the levels of the IgG control in Figure 3M. Some elaboration by the authors on this would be helpful.

As discussed in response to reviewer 1, point 3, we subtracted IgG signals from each IP in Figure 2 but not Figure 3. Figure 2 is now revised. 

4. Although the CRISP-Cas9 experiments elevate the quality of this work, the reasoning for carrying them out provided by the authors (“…it was possible that our shRNA constructs might target both Myod1 mRNA and Myog mRNA due to the high sequence homology between the factors.”) does not seem to be consistent with the results shown in Figure 1A (i.e., shMyog does not reduce Myod1 mRNA). The authors may want to reformulate the added value of these experiments.

While it still seems possible to us that the shRNA could target Myod1 mRNA and not pEF-Myod1 mRNA, the reviewer is correct that the targeting sequence of the shRNA is in both transcripts. We have updated the text accordingly.

5. More information on the antibodies used in the ChIP assays could be helpful for other investigators (e.g., product number for the anti-Histone H3 and TBP antibodies)

This information has been added to the manuscript.

---

## [Decision Letter · Decision Letter 1]

5 Jan 2021

Myogenin is required for assembly of the transcription machinery on muscle genes during skeletal muscle differentiation

PONE-D-20-22516R1

Dear Dr. Judith Davie,

We’re pleased to inform you that your manuscript has been judged scientifically suitable for publication and will be formally accepted for publication once it meets all outstanding technical requirements.

Kind regards,

Chhabi K. Govind, Ph. D.

Academic Editor

PLOS ONE

Additional Editor Comments (optional):

Reviewers' comments:

Reviewer's Responses to Questions

**Comments to the Author**

1. If the authors have adequately addressed your comments raised in a previous round of review and you feel that this manuscript is now acceptable for publication, you may indicate that here to bypass the “Comments to the Author” section, enter your conflict of interest statement in the “Confidential to Editor” section, and submit your "Accept" recommendation.

Reviewer #1: All comments have been addressed

Reviewer #2: All comments have been addressed

2. Is the manuscript technically sound, and do the data support the conclusions?

Reviewer #1: Yes

Reviewer #2: Yes

3. Has the statistical analysis been performed appropriately and rigorously? 

Reviewer #1: Yes

Reviewer #2: Yes

4. Have the authors made all data underlying the findings in their manuscript fully available?

Reviewer #1: Yes

Reviewer #2: (No Response)

5. Is the manuscript presented in an intelligible fashion and written in standard English?

Reviewer #1: Yes

Reviewer #2: Yes

6. Review Comments to the Author

Reviewer #1: The authors have addressed all of my concerns, and it seems that the revised manuscript would be acceptable.

Reviewer #2: (No Response)

7. PLOS authors have the option to publish the peer review history of their article (what does this mean?). If published, this will include your full peer review and any attached files.

Reviewer #1: No

Reviewer #2: No

---

## [Editor Report · Acceptance letter]

8 Jan 2021

PONE-D-20-22516R1 

Myogenin is required for assembly of the transcription machinery on muscle genes during skeletal muscle differentiation 

Dear Dr. Davie:

I'm pleased to inform you that your manuscript has been deemed suitable for publication in PLOS ONE. Congratulations! Your manuscript is now with our production department. 

Kind regards, 

on behalf of

Dr. Chhabi K. Govind 

Academic Editor

PLOS ONE